# Effect of *Petiveria alliacea* Extracts on Metabolism of K562 Myeloid Leukemia Cells

**DOI:** 10.3390/ijms242417418

**Published:** 2023-12-13

**Authors:** Laura Rojas, Daniel Pardo-Rodriguez, Claudia Urueña, Paola Lasso, Cindy Arévalo, Mónica P. Cala, Susana Fiorentino

**Affiliations:** 1Grupo de Inmunobiología y Biología Celular, Pontificia Universidad Javeriana, Bogotá 110211, Colombia; rojasl.a@javeriana.edu.co (L.R.); curena@javeriana.edu.co (C.U.); plasso@javeriana.edu.co (P.L.); cindy.arevalo@javeriana.edu.co (C.A.); 2Metabolomics Core Facility—MetCore, Vicepresidency for Research, Universidad de Los Andes, Bogotá 111711, Colombia; d.pardorodriguez@uniandes.edu.co

**Keywords:** myeloid leukemia, *Petiveria alliacea* aqueous extract, untargeted metabolomics, cell proliferation

## Abstract

Previously, studies have shown that leukemic cells exhibit elevated glycolytic metabolism and oxidative phosphorylation in comparison to hematopoietic stem cells. These metabolic processes play a crucial role in the growth and survival of leukemic cells. Due to the metabolic plasticity of tumor cells, the use of natural products has been proposed as a therapeutic alternative due to their ability to attack several targets in tumor cells, including those that could modulate metabolism. In this study, the potential of *Petiveria alliacea* to modulate the metabolism of K562 cell lysates was evaluated by non-targeted metabolomics. Initially, in vitro findings showed that *P. alliacea* reduces K562 cell proliferation; subsequently, alterations were observed in the endometabolome of cell lysates treated with the extract, mainly in glycolytic, phosphorylative, lipid, and amino acid metabolism. Finally, in vitro assays were performed, confirming that *P. Alliacea* extract decreased the oxygen consumption rate and intracellular ATP. These results suggest that the anti-tumor activity of the aqueous extract on the K562 cell line is attributed to the decrease in metabolites related to cell proliferation and/or growth, such as nucleotides and nucleosides, leading to cell cycle arrest. Our results provide a preliminary part of the mechanism for the anti-tumor and antiproliferative effects of *P. alliacea* on cancer.

## 1. Introduction

Despite modern advances, the therapeutic approach for acute leukemia remains a challenge, with high mortality rates and a significant percentage of patients experiencing therapeutic failure due to refractoriness or relapse [1,2]. The metabolic alterations observed in tumor cells have been proposed as one of the most significant processes in tumorigenesis [3]. It has been shown that leukemic cells, compared to hematopoietic stem cells, have an increased glycolytic metabolism and oxidative phosphorylation, which allows them to satisfy their energy demands to feed the pentose phosphate pathway (PPP) and produce the NADH and FADH_2_ necessary for the generation of ATP and citrate [4]. Additionally, leukemia cells utilize amino acid metabolism to generate crucial precursors for lipid synthesis, nucleic acids, and other cofactors [5]. These metabolic changes promote survival, proliferation, and the maintenance of an optimal redox state, which provides a protective mechanism against the stress induced by microenvironmental changes and chemotherapy [6,7]. These metabolic characteristics can be the therapeutic targets of natural products, which alone or in synergy, can alter key pathways in tumor survival and increase the response to chemotherapeutic treatment [8,9].

*Petiveria alliacea* is a perennial shrub belonging to the Petiveriaceae family, commonly known as “anamu”, and is found in the Caribbean, West Africa, and Central and South America [10]. Traditionally, infusions of its leaves and roots have been used for their hypoglycemic, anti-inflammatory, and antispasmodic properties, as well as for the management of breast cancer and leukemia [11,12]. Compounds identified in *P. alliacea* include steroids, lipids, flavonoids, triterpenes, and sulfur compounds like dibenzyl disulfide and dibenzyl trisulfide. These compounds have been attributed to cytotoxic and antiproliferative activity against various tumor cell lines [13,14,15].

In previous reports, our research group demonstrated the cytotoxic activity of the ethyl acetate (EtOAc) fraction from *P. alliacea* on melanoma, breast, and leukemia tumor cell lines. This activity is associated with apoptosis via both mitochondrial-dependent and independent routes, affecting cell morphology and altering the proteins involved in energy metabolism, such as glucose phosphate isomerase, phosphoglycerate dehydrogenase, pyruvate kinase, glyceraldehyde-3-phosphate dehydrogenase, and ATP synthase, as well as those implied in cell proliferation. This fraction also decreases the β-F1-ATPase in murine (4T1) and human breast cell lines, leading to a rapid decrease in intracellular ATP [16,17]. Additionally, it induces cell cycle arrest in K562, Mel Ret, and A375 tumor cells [15,16,17,18]. Furthermore, other research has demonstrated that myeloid leukemia cells are more sensitive to mitochondrial damage than other tumor models, partly because they have a reduced maximum respiratory capacity, making them more responsive to metabolic modulators [19,20].

Given that indigenous and peasant communities traditionally use a water infusion of *P. alliacea* called “Esperanza” for the treatment of cancer, this research aimed to thoroughly evaluate the metabolic alterations induced by this aqueous extract, previously characterized [21], on K562 leukemic tumor cells. We employed a multi-platform untargeted metabolomics approach and in vitro biological activities to investigate the potential mechanisms of action associated with its anti-tumor activity.

## 2. Results

### 2.1. The Esperanza Extract Reduces the Cell Proliferation Rate without Inducing Apoptosis

The cytotoxic activity of the Esperanza extract was assessed using the MTT colorimetric technique. Both tumor cells (K562) and non-tumor cells (3T3 and HPdlF) were treated with different concentrations of the extract (500 μg/mL–3.90 μg/mL) for 48 h. In contrast to the chemotherapeutic drug doxorubicin, Esperanza extract did not present a cytotoxic effect on K562, 3T3, or HPdLF cells at the concentrations evaluated (Figure 1A). The proliferative capacity of K562 cells was significantly inhibited in a dose-dependent manner after treatment with the Esperanza extract (Figure 1B), with the population doubling time (PDT) being ten times greater for the highest concentration (155 μg/mL). The induction of apoptosis and necrosis in cell populations was determined using annexin V and propidium iodide labeling to confirm that the extract acts particularly as a cytostatic agent. The cells treated with the Esperanza extract showed a low frequency of apoptotic death at 24 h (8.04%), 48 h (8.93%), 72 h (12.06%), and 96 h (13.38%) compared to the positive control doxorubicin (Appendix A, Figure 1C).

### 2.2. Alterations in the Endometabolome of the K562 Cell Line Treated with Esperanza Extract

Using a multi-platform untargeted metabolomic approach, the metabolic changes linked to the Esperanza extract administration in K562 cells were evaluated in order to identify a broad spectrum of changed metabolites. Initially, principal component analysis (PCA) was employed to assess the performance of the various systems. A close examination of the model revealed a distinct clustering of cell lysates from the quality control (QC) group samples (orange dots) treated with Esperanza and PBS (gray dots) on all analytical platforms (Appendix A) used, indicating the preservation of biological variation.

Furthermore, to maximize the differences between the groups of cell lysates treated with Esperanza extract and the group of cell lysates treated with PBS and to identify the molecular features with the greatest influence on separation, an orthogonal partial least squares discriminant analysis (OPLS-DA) was employed. The OPLS-DA score plot (Figure 2A,B) demonstrated clear discrimination between cell lysates treated with Esperanza (red dots) and those treated with PBS (blue dots), with R^2^ values of 0.999 and Q^2^ values of 0.966 for LC-QTOF-MS and R^2^ values of 0.984 and Q^2^ values of 0.937 for GC-QTOF-MS, suggesting acceptable goodness of fit and prediction [22]. To evaluate the possible overfitting in the supervised OPLS-DA models, a cross-validation analysis (K-fold CV-ANOVA) and a random permutation analysis were performed. For all analytical platforms, the K-Fold CV-ANOVA analysis yielded significant pCV values *(*<0.05), and in the permutation analysis, the y-intercepts of the Q^2^ distributions remained systematically below zero (Appendix A). These results indicated that the models did not suffer overfitting [23].

Through a combination of multivariate analysis (MVA) with variable importance in projection (VIP > 1 with JK) and univariate analysis (UVA) (*p* < 0.05), a total of 48 altered metabolites were identified in the treated cell cultures (Appendix A, Figure 2C), of which 47.92% were identified by LC-MS and 52.08% by GC-MS (Appendix A). The consolidated analysis across both analytical platforms revealed that 47.92% of the metabolites were elevated, with chemical families such as glycerophospholipids (26.09%) and fatty acids (17.39%) predominantly increased. On the other hand, 52.08% of the compounds exhibited decreasing trends, with the chemical family of nucleosides, nucleotides, and analogs (24.0%) being the most representative. Other chemical families such as indoles, oxygenated compounds, and carboxylic acids did not show consistent trends throughout the comparison.

Heatmaps were used to examine the set of changed metabolites between treated and untreated cells. In this case, the range of colors from green to red indicates decreased and increased levels of metabolites, respectively (Figure 3A). The results show that cells treated with Esperanza extract exhibited increased levels (metabolites in the red color range) of lipid-type metabolites (hydroxyoctadecatrienoylcarnitine, hydroxyoctenoylcarnitine, acetylcarnitine, PE 20:4, PC18:3, PC18:3, PI 24:0, PG O-39:0, PS 2-OMe,14Me-15:0, and sphingosine), as well as an increase in some sugars (sorbose, phosphogluconic acid, glucose, and mannitol) and amino acids (serine, proline, glycine, methionine, and lactoyl leucine). This pattern was also seen in other carboxylic acids, such as glutathione and oxalic acid. On the other hand, decreased metabolites (metabolites in the green color range) were observed in the family of nucleosides and nucleotides compounds (guanosine monophosphate, NADH, thioinosine monophosphate, methylthioadenosine, inosinic acid, and adenosine monophosphate), carboxylic acids (alanine, glutamine, tyrosine, leucine, norleucine, fumaric acid, pentanedioic acid/glutaric acid, glutamic acid, dicarboxyethyl glutathione, oxoproline/pyroglutamic acid, methyl pyruvic acid, and malic acid), and other metabolites such as gluconic acid, myo-inositol, pantothenate, tryptophanol, propionylcarnitine, uracil, and hypoxanthine.

To further investigate the metabolic changes brought about by Esperanza extract, treated samples of cellular lysates were contrasted with controls (PBS). The MetaboAnalyst 5.0 server was used to import the previously identified metabolites and perform a thorough metabolic enrichment analysis. Figure 3B shows a summary of the modified metabolic pathways. Higher levels of enrichment are indicated by larger dots; as a result, the enrichment factor is represented by the x-axis, and the statistical significance of the enrichment is indicated by the y-axis, which represents the −log10 (*p*-value) [24]. Following treatment with Esperanza extract, the metabolic pathways that had the greatest impact (highlighted in various shades of red) and the greatest number of metabolites involved in cellular lysates were primarily linked to RNA biosynthesis, which was related to the involvement of metabolites like glutamine, glycine, serine, methionine, alanine, leucine, tyrosine, and glutamate. Furthermore, there was a notable impact on the pathways linked to the metabolism of glutathione, glycine, serine, cysteine, threonine, and glyoxylate as well as dicarboxylate. Although they were not as significant, other pathways also showed some influence.

To confirm whether the metabolic changes exerted by Esperanza in K562 cells have an impact on glycolysis and mitochondrial function. We assessed the oxygen consumption rate (OCR), a measure of mitochondrial function, and the medium acidification rate (ECAR), an indirect measure of glycolysis. Esperanza extract induced a significant reduction in the OCR (Figure 4A) and a slight increase in the ECAR (Figure 4B). The maximum glycolytic capacity (MGC) was then demonstrated using oligomycin, and the maximum respiratory capacity (MRC) was determined using carbonyl cyanide-p-trifluoromethoxyphenylhydrazone (FCCP). This method aims to assess the degree of reversibility of the modifications made to energy metabolism. It was observed that the maximum glycolytic capacity (MGC) was approximately 9.2 ± 2.1 µs (lifetime), and the application of Esperanza significantly reduced it to 3 ± 1.7 µs (lifetime) (Figure 4C). Notably, this effect differed from the maximal respiratory capacity (MRC), which remained unaffected by the extract. At the basal level, the MRC measured 93.0 ± 17.9 µs (lifetime), and after treatment with Esperanza, it registered at 89.4 ± 12.5 µs (lifetime) (Figure 4D). The energy map presented in Figure 4E provides evidence that untreated K562 cells exhibit high metabolic activity, while treatment with Esperanza induces a shift toward a glycolytic phenotype.

On the contrary, the evaluation of intracellular ATP levels revealed a notable decrease in cells treated with Esperanza compared to untreated cells (Figure 4F). Interestingly, these findings align with the effects observed with previously identified compounds in *P. alliacea*, such as dibenzyl disulfide (DBS), dibenzyl trisulfide (DTS), and myricetin. This collective set of results underscores that Esperanza induces mitochondrial damage, potentially of an irreversible nature, thereby impacting cellular energy production. Given the metabolic observations in the cell lysates, the modest increase in the extracellular acidification rate (ECAR) is attributed to potential non-glycolytic acidification.

## 3. Discussion

The metabolic alterations observed in tumor cells have been proposed as one of the most critical processes in tumorigenesis [3]. As a therapeutic alternative, the use of natural products has been suggested, with reported anti-tumor activity, cytotoxicity, and the ability to modulate tumor metabolism [8]. *P. alliacea* has been used for many years to treat cancer, including leukemia [10,25]. Historically, Native American communities have used infusions made from its leaves and roots to treat rheumatism, fever, intestinal parasites, and cancer [12,26]. The mechanisms of action described to date have involved alterations in the cytoskeleton, cell cycle arrest, induction of apoptosis through caspase 3 activation, and DNA fragmentation without mitochondrial membrane depolarization in various cellular models [15,16,18]. On a metabolic level, *P. alliacea* induces changes in the cellular expression of several glycolytic enzymes, a decrease in glycolysis, lactate production, and mitochondrial respiration, and a reduction in β-F1-ATPase, ultimately leading to ATP deficiency [16,17]. It is interesting to note that despite not exhibiting significant cytotoxic activity against leukemia cell lines, it does noticeably decrease cell replication. This effect may be related to alterations in tumor metabolism, the reduction in glycolysis, and/or the previously described mitochondrial respiration alterations [27].

Among the findings, an increase in intracellular glucose was noted in cell lysates treated with the Esperanza extract. Notably, it has been documented that the inhibition of glycolysis can lead to elevated glucose levels [28,29,30]. We previously showed a reduction in glycolysis in a breast cancer cellular model (4T1) and an in vivo leukemia model [18,21]. This reduction can be attributed to the primary compounds previously reported in the extract, such as isoleucine and valine, which have shown the ability to decrease proliferation and tumor growth by upregulating the tumor suppressor gene PTEN. PTEN acts as an inhibitor of phosphoglycerate kinase 1, a crucial enzyme in glycolysis [31]. Furthermore, other compounds identified within the extract, including aspartic acid and glutamic acid, may explain its activity on metabolism through the inhibition of Akt signaling pathways [32]. Despite the fact that *P. alliacea* has been shown in other studies to have an inhibitory effect on glycolysis, we observed a rise in the ECAR, leading us to believe that acidification is mediated by mechanisms other than glycolysis. The TCA cycle and intracellular glycogen destruction, or glycogenolysis, are the two main proton sources that may contribute to non-glycolytic acidification [33]. Later, this hypothesis will need to be verified.

On the other hand, cellular K562 lysates treated with Esperanza exhibited a decrease in tyrosine, glutamine, and glutamate, which are key precursors in the tricarboxylic acid cycle (TCA) (Figure 5). A reduction in these precursors leads to mitochondrial dysfunction, resulting in decreased energy production and reduced oxidative capacity, as observed in the decreased OCR [34]. Previously, the role of *P. alliacea* in mitochondrial function was demonstrated, mediated by an ethyl acetate fraction which, in vitro, induces apoptosis in 4T1 cells, activates caspase 3, and causes DNA fragmentation [16,18]. It is also worth noting that an additional impact on mitochondria possibly induced by malic acid has been previously described with the Esperanza extract [21]. Malic acid is known to inhibit succinate dehydrogenase, an enzyme that plays a central role in the TCA cycle and as part of complex II in the electron transport chain [35].

The observed decrease in glutamine in response to treatment could potentially result from the reduction and/or inhibition of the SCL1A5 receptor [36]. Although the activity of compounds present in *P. alliacea* on this receptor has not been described, berberine, the principal compound in many Chinese medicinal herbs, has been shown to decrease glutamine uptake following the inhibition of SCL1A5 [37]. The negative regulation of glutamine consumption directly impacts the TCA cycle, reducing its activity and consequently inhibiting OXPHOS. This could be an intriguing therapeutic target that has not been previously highlighted for *P. alliacea*.

Each of the results described above suggests that leukemia cells undergo a series of metabolic changes in response to Esperanza treatment, particularly in glycolysis and at the mitochondrial level. These changes could be the cause of decreased cell proliferation and intracellular ATP levels. Interestingly, we observed a decrease in the nucleotides adenine and guanosine, which are important precursors in DNA and RNA synthesis [38]. Additionally, there was an increase in lipids, especially acyl carnitines, and glycerophospholipids. An accumulation of glycerophospholipids can trigger lipotoxicity and, consequently, cause mitochondrial dysfunction [39]. On the other hand, an increase in carnitines could indicate two scenarios: mitochondrial dysfunction due to β-oxidation inhibition or, conversely, it could be a consequence of accelerated fatty acid oxidation in the mitochondria once glycolysis is inhibited [40,41]. Previous studies have demonstrated that increased carnitine-dependent fatty acid uptake into mitochondria can induce mitochondrial dysfunction [41].

In addition to the primary compounds, the secondary compounds previously characterized in the extract [21] also exhibit various anti-tumor activities: ferulic acid induces apoptosis and inhibits tumor necrosis factor-α and Akt/mTOR signaling in leukemic cells [42]; cinnamic acid induces apoptotic cell death and alteration of the cytoskeleton in human melanoma cells [43]; and quinic acid, an antioxidant, exhibits anti-tumor activity through the induction of apoptosis and decreased angiogenesis in breast cancer cells [44]. Finally, a widely studied compound is gallic acid, a phenolic compound that modulates genes involved in the cell cycle, metastasis, and angiogenesis, leading to the inhibition of tumor cell growth; it also inhibits key signaling pathways for proliferation such as NF-KB and Akt [45,46,47].

In summary, the aqueous extract of *P. alliacea* induces multiple metabolic alterations in K562 cells, primarily leading to a decrease in metabolites related to the cell cycle, such as nucleosides, nucleotides, and certain amino acids, as well as the accumulation of carbohydrates, thereby promoting cell cycle inhibition. It should be noted that this set of results should be further confirmed by metabolic flux analysis (fluxomics) or by the use of isotopic tracers, such as ^13^C-labeled substrates, together with analytical techniques such as mass spectrometry and nuclear magnetic resonance which will allow us to understand and optimize the possible mechanism of action exerted by *P. alliacea* on tumor cells and additionally provide detailed information on the distributions of metabolic flux.

## 4. Materials and Methods

### 4.1. Plant Material and Extraction

The aqueous extract of *P. alliacea* was obtained and previously characterized by our group as reported [21]. Briefly, the leaves of *Petiveria alliacea* were collected, washed, and subjected to an extraction procedure by infusion with boiling water. Subsequently, this extract was subjected to freeze-drying. Chromatographic profiling was performed using ultra-performance liquid chromatography coupled with photodiode array detection (UPLC-PDA) at a wavelength of 254 nm. The production of the Esperanza extract was carried out under good manufacturing practice (GMP) conditions at Labfarve Laboratories with subsequent physicochemical and microbiological certification before starting the experiment. 

### 4.2. In Vitro Cytotoxicity and Proliferation Assays

The K562 leukemic cell line (chronic myeloid leukemia), the 3T3 non-tumor cells (murine fibroblasts). and the human periodontal ligament fibroblasts (HPdLF) were cultured in RPMI supplemented with heat-inactivated bovine fetal serum (10%) (Eurobio, Toulouse, France), 2 mM glutamine, 100 U/mL penicillin–streptomycin (Gibco), 0.01 M Hepes (Gibco), and 1 mM sodium pyruvate (Gibco, Waltham, MA, USA). The cells were cultured at a concentration of 1 × 10^6^ cells under a humidified environment at 37 °C and 5% CO_2_. This methodology was employed to maintain the optimal conditions for cell growth and viability [48]. The Esperanza extract’s cytotoxic effect on tumor cells was evaluated using a methylthiazol tetrazolium (MTT) assay (Sigma-Aldrich, Saint Louis, MO, USA) as previously reported [49]. The 50% inhibitory concentration value (IC_50_) was calculated with nonlinear regression curve fitting using GraphPad Prism version 8.0.1 (GraphPad Software, San Diego, CA, USA). Experiments were performed in triplicate on three independent experiments and the results were expressed as the mean ± SEM.

For the proliferation assay, 1 × 10^5^ K562 cells were seeded on 6-well plates and subsequently treated with 15 and 31 µg/mL of the Esperanza extract, or PBS (0.2%) as vehicle control, for 12, 48, 72, and 96 h. After incubation, the cells were counted in a Neubauer chamber with trypan blue. The exponential (Malthusian) growth formula Y = Y0 × exp(k × x) of the Software GraphPad Prism version 8.0.1 for Mac OS X was used to calculate the doubling time. The experiment was repeated three times in triplicate, and the results were expressed as the mean ± SEM. 

### 4.3. Cell Death Evaluation

A total of 2 × 10^5^ K562 cells were treated with 155 µg/mL of the Esperanza extract for 24 and 48 h at 37 °C. DMSO (<1%) or H_2_O was used as a negative control and doxorubicin (0.40 µM) as a positive control. After incubation, cells were washed with PBS and labeled with annexin V (Molecular Probes, Invitrogen Corp, Carlsbad, CA, USA) and propidium iodide (PI) (Sigma, Saint Louis, MO, USA) [50]. A total of 10,000 events were acquired on a FACSAria II-U flow cytometer (Becton Dickinson, BD, Franklin Lakes, NJ, USA). The results were subsequently analyzed using FlowJo v10.8.1 software (BD Life Sciences, Franklin Lakes, NJ, USA). Double-negative cells were considered intact, whereas double-positive cells were considered in late apoptosis/necrotic cells. Annexin V^+^/PI^−^ cells were presumably in early apoptosis and the Annexin V^−^/PI^+^ were considered necrotic cells.

### 4.4. OCR and ECAR Evaluation

The Agilent MitoXpress and pH-Xtra assays were used to measure the oxygen consumption rate (OCR) and the extracellular acidification rate (ECAR) in a Cytation5 Reader (BioTek, Winooski, VT, USA), following the manufacturer’s instructions. The ECAR and OCR were simultaneously measured in basal conditions and after the subsequent treatment. Briefly, K562 cells were seeded in a 96-well plate at 1.25 × 10^4^ per well and then treated with 155 μg/mL of Esperanza for 6 h. Before carrying out the measurement, the controls were added. As a control for the ECAR, 50 mM 2-DG (2-deoxy-d-glucose >98%, Sigma-Aldrich, Burlington, MA, USA) was used, and for the OCR 1, μM AntiA (Antimycin A from Streptomyces sp, Sigma-Aldrich, St. Louis, MO, USA) were used. Dual-read TR-F and a subsequent lifetime calculation allowed for the measurement of the rate of fluorescence decay of the MitoXpress Xtra and pH-Xtra reagents. The dual intensity readings were used to calculate the corresponding lifetime (µs) using the following transformation: lifetime (µs)[τ] =(D2 − D1)/ln(IW1/IW2), where IW1/ and IW2 represent the two (dual) measurement windows and D1 and D2 represent the delay time prior to measurement of W1 and W2, respectively. 

### 4.5. ATP Determination

To evaluate intracellular ATP, 2.5 × 10^5^ K562 cells were seeded on a 12-well plate and treated with 155 μg/mL of Esperanza and H_2_O (negative control, 0.02%) for 6 h. Intracellular ATP was measured with an ATP Bioluminescence Assay Kit HS II (Roche, Mannheim, Germany) following the manufacturer’s instructions. For assessment of the chemoluminescent signal, the plates were read in a BioTek Cytation 5 Cell Imaging Multimode Reader (Winooski, VT, USA). Experiments were performed in triplicate on two independent experiments and the results were expressed as the mean ± SEM.

### 4.6. Untargeted Metabolomics Analysis

#### 4.6.1. Sample Preparation 

The cells (1 × 10^6^) were treated with the Esperanza extract at 155 µg/mL, or PBS as a negative control, for 12 h. After incubation, the cells were washed three times with PBS at 4 °C, frozen in liquid nitrogen, and stored at −80 °C until further processing. Six independent biological replicates were evaluated for each treatment.

To isolate the metabolites from both the treated and control samples, a solution of 500 µL MeOH-water (4:1 *v*/*v*) was added to each sample. After being centrifuged at 15,700 g and 4 °C for 20 min, the samples underwent freezing and thawing cycles for lysis. Subsequently, the samples were passed through 0.22 µm filters. Following this, the samples were transferred into vials for liquid chromatography (LC) and gas chromatography (GC) coupled to mass spectrometry with a time-of-flight analyzer (QTOF/MS).

#### 4.6.2. Untargeted Metabolomics by LC-QTOF-MS

The samples were analyzed using an Agilent Technologies 1260 Liquid Chromatography system coupled to a Q-TOF 6545 time-of-flight mass spectrometer with electrospray ionization. A total of 5 µL of the extracts from each sample were injected into a C18 column (100 mm × 3.0 mm, 2.7 µm) and eluted using a gradient comprising of 0.1% (*v*/*v*) formic acid in Milli-Q water (Phase A) and acetonitrile (Phase B) at a constant flow rate of 0.3 mL/min and a temperature of 40 °C. The initial gradient consisted of 5% of Phase B and was maintained for 10 min, after which it was gradually increased to 95% B over a period of 12 min and maintained at this proportion for 2 min. The gradient was then reduced to 5% of Phase B and held for five additional minutes for equipment re-equilibration. Full scan mass spectrometry acquisition was carried out from 70 to 1100 *m*/*z* in positive electrospray ionization (ESI) mode. Throughout the analysis, two reference masses were used for mass correction, *m*/*z* 121.0509 [C_5_H_4_N_4_ + H]^+^ and *m*/*z* 922.0098 [C_18_H_18_O_6_N_3_P_3_F_24_ + H]^+^.

#### 4.6.3. Untargeted Metabolomics by GC-QTOF-MS

A total of 100 µL of metabolite extracts were evaporated under vacuum using a SpeedVac at 35 °C for 2 h. Next, 20 µL of O-methoxymine in pyridine (15 mg/mL) was added, vortexed for 5 min, and incubated in the dark at room temperature for 16 h. The sililation process was conducted by adding 20 µL of N,Obis(trimethylsilyl)trifluoroacetamide with 1% Trimethysilyl, followed by vortexing for 5 min and incubating at 70 °C for 1 h. Subsequently, the samples were allowed to cool to room temperature for 30 min, and 100 µL of methyl stearate in heptane as an internal standard (10 mg/L) was added and vortexed for 10 min. 

For data acquisition, an Agilent Technologies 7890B gas chromatograph coupled to an Agilent Technologies GC/Q-TOF 7250 time-of-flight mass selective detector was employed. The system was equipped with a split/splitless injection port (250 °C, split ratio 10) and an Agilent Technologies 7693A automatic injector. The electron ionization (EI) source was set to 70 eV. A column from Agilent Technologies J&W HP-5MS (30 m, 0.25 mm, 0.25 µm) was utilized, and helium was used as the carrier gas at a constant flow rate of 0.7 mL/min. The oven temperature was programmed to increase from 60 °C (1 min) to 325 °C (10 min) at a rate of 10 °C/min. The transfer line temperature to the detector, the source filament temperature, and the quadrupole temperature were maintained at 280 °C, 230 °C, and 150 °C, respectively. The acquisition was conducted in the range of 50 to 600 *m*/*z* at a speed of 5 spectra/min.

#### 4.6.4. Quality Control Samples 

Quality control samples (QC) samples were produced by combining the metabolic extract from each sample in equal volumes. After that, the preparation and examination of the QC samples were conducted using the methods described above for each analytical platform. To evaluate the consistency of sample preparation and the reliability of the analytical system employed, numerous QC injections were performed until the analytical system had stabilized. Following that, the QC samples were analyzed after every six randomly chosen sample injections.

### 4.7. Data Processing and Analysis

The LC-QTOF-MS data were processed using the Agilent MassHunter Profinder B.10.0 software to deconvolute, align, and integrate the data using recursive and molecular feature extraction algorithms. The GC-QTOF-MS data were treated using the Agilent MassHunter Unknowns Analysis B.10.0 software and the Fiehn and NIST libraries. A retention time alignment was then performed in the Agilent Mass Profiler Professional B.15.0 software, and the results were exported to the Agilent MassHunter Quantitative B.10.0 software for data integration. The resulting GC-QTOF-MS and LC-QTOF-MS data were manually inspected and then filtered by presence and reproducibility, retaining only metabolites present in 100% of samples in the same group, with a coefficient of variation of less than 20% in the QC. 

The identification of molecular features showing statistically significant differences between the experimental groups was performed using both univariate (UVA) and multivariate (MVA) statistical analysis. For the UVA analysis, the *p*-value was calculated using nonparametric tests (Mann–Whitney U test) with the Benjamini–Hochberg False Discovery Rate post hoc correction (FDR) using MatLab 2019b, Mathworks, Inc., Natick, MA, USA. During the MVA analysis, an unsupervised principal component analysis (PCA) was conducted to observe the unsupervised distribution of the samples analyzed. Subsequently, supervised orthogonal partial least squares discriminant analysis (OPLS-DA) models were performed to select the molecular features responsible for the separation between the groups. The MVA analysis was carried out using the SIMCA-P+16.0 software (Umetrics, Vasterbottens Lan, Sweden). The selected molecular characteristics showed statistical significance, meeting at least one of the following criteria: (1) UVA: *p*-value < 0.05, and (2) MVA: variance important in projection (VIP) > 1.

### 4.8. Annotation of Statistically Significant Molecular Features

The Fiehn version 2013 libraries, MassHunter Personal Compound Database, and Library Manager Software B.08.00 were used to identify the metabolites discovered by GC-QTOF-MS analysis. The identification of metabolites was carried out based on a four-level confidence system for high-resolution mass spectrometry analysis following the parameters in reference [23]. The metabolites were annotated by matching the exact observed mass of each compound with the available *m*/*z* values in the online databases at METLIN (http://metlin.scripps.edu (accessed on 16 August 2023)), KEGG (http://genome.jp/kegg (accessed on 16 August 2023)), LIPIDMAPS (http://lipidMAPS.org (accessed on 16 August 2023)), and HMBD (http://hmdb.ca (accessed on 16 August 2023)). The identity of the metabolite was confirmed through MS/MS data analysis; this analysis was performed using the Agilent Lipid Annotator, MS-DIAL 4.80 (http://prime.psc.riken.jp/compms/msdial/main.html (accessed on 29 August 2023)), and CFM-ID 4.0 (https://cfmid.wishartlab.com/ (accessed on 29 August 2023)) for in silico mass spectral fragmentation. 

### 4.9. Altered Metabolite Pathway Mapping

An analysis of the metabolic pathways affected in the treated samples was performed using the “Enrichment Analysis” tool available on the MetaboAnalyst 5.0 server (http://www.metaboanalyst.ca/ (accessed on 14 September 2023)). The enrichment test used a linear model to calculate a “Q-stat” for each set of metabolites and Q-covariance values. To obtain a comprehensive analysis of the metabolic pathways, the “Homo sapiens” (KEGG) pathway library was selected.

## 5. Conclusions

The aqueous extract of *Petiveria alliacea*, known as “Esperanza”, diminishes the proliferation of chronic myeloid leukemia K562 cells through mechanisms associated with oxidative and energy metabolism. The observed metabolic alterations suggest a decrease in glycolytic intermediates and precursors of the tricarboxylic acid cycle, with a significant reduction in the synthesis of nucleotides and essential amino acids for cell proliferation. In vitro biological activity data show that Esperanza decreases the oxygen consumption rate, potentially irreversibly, impairing the ability of cells to generate energy. This biological activity may be attributed to the presence of compounds such as malic acid, ferulic acid, cinnamic acid, quinic acid, and gallic acid, previously described in the extract. This partly explains the anti-tumor activity of anamu however, although it is known that the tumor energy metabolism is more accelerated than that found in cells with low replication rates, it remains to be explained in detail what makes *P. Alliacea’s* biological activity specific to some tumor types.

## Figures and Tables

**Figure 1 ijms-24-17418-f001:**
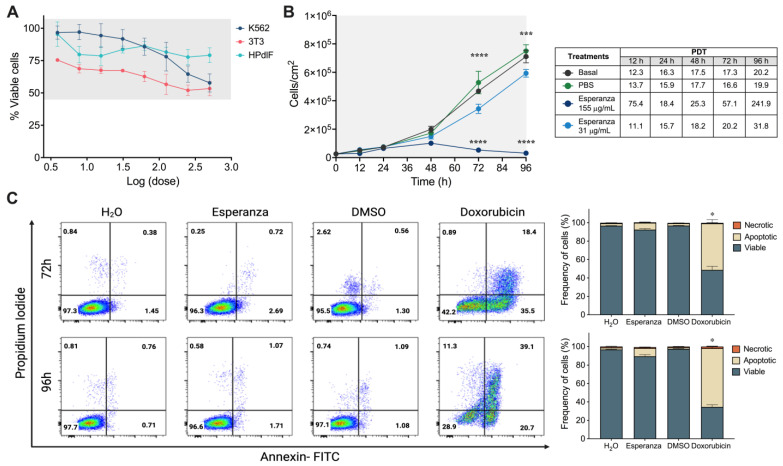
Assessment of the antiproliferative effect of Esperanza extract on tumor cell lines. (**A**) Dose-response viability curve of K562, 3T3, and HPdLF cell lines treated with Esperanza extract. (**B**) Growth kinetics of K562 tumor cells after treatment with Esperanza extract. (**C**) Cell death induction and frequency of live cells (Annexin V^−^, PI^−^), apoptotic cells (Annexin V^+^, PI^−^ and Annexin V^+^, PI^+^), and necrotic cells (Annexin V^+^, PI^+^) in K562 cells. The data from three independent experiments are presented. The *p*-values indicate a statistically significant difference between the compared treatments, * *p <* 0.05; *** *p <* 0.001; and **** *p <* 0.0001. In (**C**), significant differences were observed in the percentage of apoptotic cells after the treatment with doxorubicin compared to the rest of the treatments (yellow bars).

**Figure 2 ijms-24-17418-f002:**
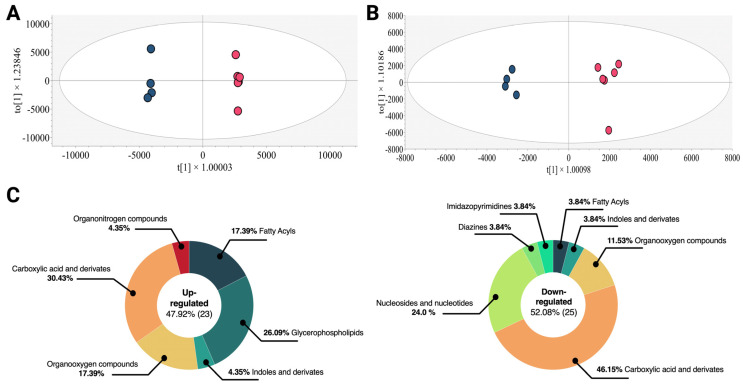
OPLS-DA models of the metabolic analysis of treated cells and control group and altered metabolites in the treated cells according to their chemical classes. (**A**) LC-QTOF-MS (+) R^2^: 0.825, Q^2^: 0.678. (**B**) GC-MS (+) R^2^: 0.540, Q^2^: 0.199. Blue dots represent cell lysates treated with PBS and red dots represent cell lysates treated with Esperanza. (**C**) Altered metabolites in K562 treated cells. Left panel, up-regulated metabolites. Right panel, down-regulated metabolites. The chemical classes are shown according to the color code.

**Figure 3 ijms-24-17418-f003:**
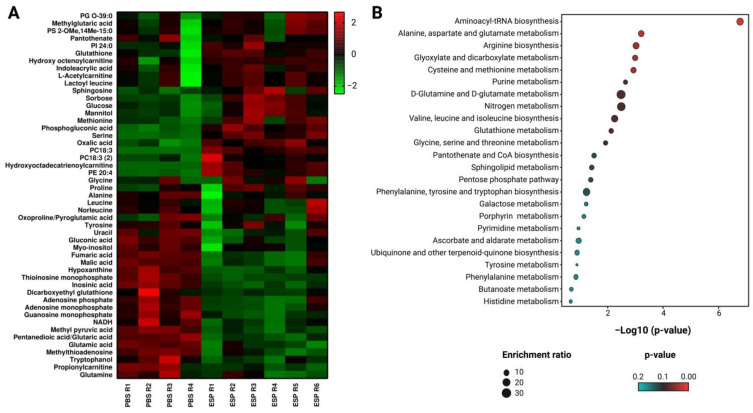
(**A**) Heat map of metabolites with a statistically significant variation between K562 cells treated with Esperanza extract and the control group. The rows correspond to each altered metabolite identified; the columns correspond to the analyzed samples. The level of variation is indicated on the right side on a color intensity scale representing relative abundance, where red colors denote metabolite increase and green colors denote metabolite decrease. (**B**) Enrichment analysis of the altered pathways in K562 cells treated with Esperanza extract. The significance of pathway alteration is indicated according to the color scale, dots in cyan colors represent non-significant alterations (*p* > 0.05), and red represents significant alterations (*p* < 0.05).

**Figure 4 ijms-24-17418-f004:**
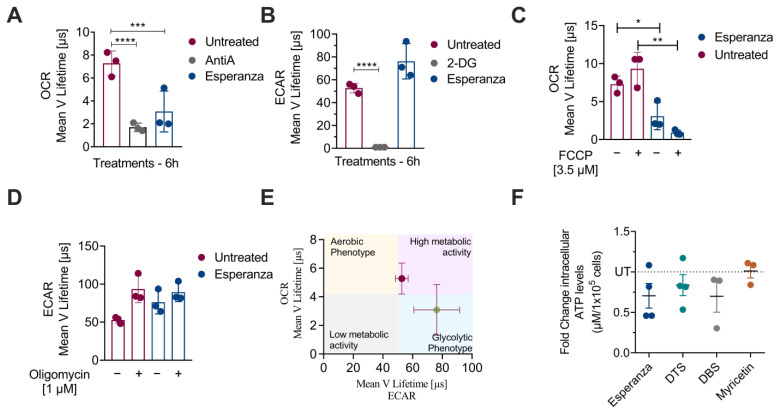
(**A**) Oxygen consumption rate (OCR) of K562 cells treated with Esperanza extract and the untreated group. (**B**) Medium acidification rate (ECAR) of K562 cells treated with Esperanza extract and the untreated group. (**C**) The maximum glycolytic capacity (MGC). (**D**) The maximal respiratory capacity (MRC). (**E**) Energy map of K562 cells treated with Esperanza extract (green dot) and the untreated group (fuchsia dot). (**F**) Intracellular ATP levels of K562 cells treated with Esperanza and compounds (DTS, DBS, and myricetin). The data from three independent experiments are presented. The *p*-values indicate a statistically significant difference between the compared treatments. * *p* < 0.05; ** *p* < 0.01; *** *p* < 0.001; and ***** p* < 0.0001.

**Figure 5 ijms-24-17418-f005:**
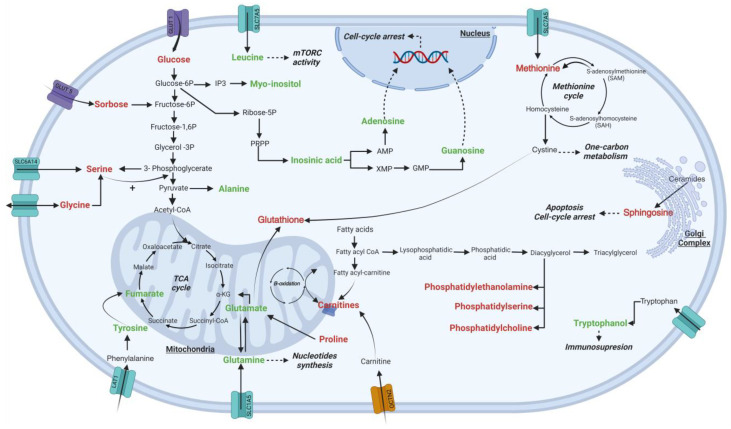
Changes in the metabolic pathways of K562 cells after treatment with the Esperanza extract. Metabolites with decreasing and increasing trends in treated cells are indicated in green and red, respectively.

## Data Availability

The data presented in this study are available in this article (and Appendix A).

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
