# Peer review of "Effect of Petiveria alliacea Extracts on Metabolism of K562 Myeloid Leukemia Cells"

_ijms, 2023, doi:10.3390/ijms242417418_

Round 1
Reviewer 1 Report
Comments and Suggestions for Authors
A study by Laura Rojas Fonseca et al investigates effect of Petiveria alliacea extracts on metabolism of K562 cells. The Authors performed extensive metabolic analysis, and provided interesting data that however should be extended to substantiate the conclusions providing that a single cell line was used in this study.
Specific comments:
1. Abstract lacks conclusions of the study.
2. Fig. 1C - Annexin V/PI staining should be performed after 72 or 96 hours to support the conclusion that 'The Esperanza extract reduces the cell proliferation rate without inducing apoptosis'. As show in Fig. 1B - after 48h, there is no substantial changes in a number of viable cells, which is however significantly reduced later.
3. Results obtained using untargeted metabolomics analysis should be validated using at least one additional method to substantiate the quality of the manuscript.
Comments on the Quality of English Languageminor revision
Author Response
The responses to each of the observations made by the reviewers are in blue, immediately after each observation. Regarding the corrected version of the manuscript (Rojas et al ID ijms-2697444) it is sent in an attached file with the changes highlighted (yellow).
Reviewer 1
A study by Laura Rojas Fonseca et al investigates effect of Petiveria alliacea extracts on metabolism of K562 cells. The Authors performed extensive metabolic analysis, and provided interesting data that however should be extended to substantiate the conclusions providing that a single cell line was used in this study.
Specific comments:
- Abstract lacks conclusions of the study.
Response: We apologize for the error; the pertinent correction is made.
- 1C - Annexin V/PI staining should be performed after 72 or 96 hours to support the conclusion that 'The Esperanza extract reduces the cell proliferation rate without inducing apoptosis'. As show in Fig. 1B - after 48h, there is no substantial changes in a number of viable cells, which is however significantly reduced later.
Response: In response to the reviewer's suggestion, we incorporated the evaluation of Annexin V/PI cell death at 72 and 96 hours (Figure 1C), confirming that the "Esperanza extract reduces the rate of cell proliferation without inducing apoptosis". Furthermore, we have included the assessment of Annexin V/PI cell death at 24 and 48 hours in the supplementary material.
- Results obtained using untargeted metabolomics analysis should be validated using at least one additional method to substantiate the quality of the manuscript.
Response: In accordance with the reviewer's suggestion, we incorporated experiments to evaluate the oxygen consumption rate (OCR) (Figure 4A-C), extracellular acidification rate (ECAR) (Figure 4 D-E), and intracellular ATP levels using the ATP Bioluminescence Assay Kit HS II (Roche, Mannheim, Germany) (Figure 4F). These assessments were conducted on the Cytation5 Reader (BioTek, Vermont, USA).
Reviewer 2 Report
Comments and Suggestions for Authors
In this study, the authors reported that Petiveria alliacea extracts reduces K562 cell proliferation by affecting the metabolism. The authors reported that these findings are due to the effect on glycolytic, phosphorylative, lipid, and amino acid metabolism pathways.
The authors utilized metabolomics as a new approaches, but some concerns are raised.
1) Since the authors hypotheized the enerygy metabolism associated with reduced proliferation such as reduced glycolysis. I would suggest confirm these finding by other functional assays that measure glycolytic and mito ATP such as Seahorse.
2) Language editing is required.
3) Did the author include poisitve control (as anti-cancer) in the metabolomics assays? Did they notice the same pathways?
Comments on the Quality of English LanguageLanguage editing is required
Author Response
The responses to each of the observations made by the reviewers are in blue, immediately after each observation. Regarding the corrected version of the manuscript (Rojas et al ID ijms-2697444) it is sent in an attached file with the changes highlighted (yellow).
Reviewer 2
In this study, the authors reported that Petiveria alliacea extracts reduces K562 cell proliferation by affecting the metabolism. The authors reported that these findings are due to the effect on glycolytic, phosphorylative, lipid, and amino acid metabolism pathways.
The authors utilized metabolomics as a new approaches, but some concerns are raised.
1. Since the authors hypotheized the enerygy metabolism associated with reduced proliferation such as reduced glycolysis. I would suggest confirm these finding by other functional assays that measure glycolytic and mito ATP such as Seahorse.
Response: In accordance with the reviewer's suggestion, we incorporated experiments to evaluate the oxygen consumption rate (OCR) (Figure 4A-C), extracellular acidification rate (ECAR) (Figure 4 D-E), and intracellular ATP levels using the ATP Bioluminescence Assay Kit HS II (Roche, Mannheim, Germany) (Figure 4F). These assessments were conducted on the Cytation5 Reader (BioTek, Vermont, USA).
2. Language editing is required.
Response: Consistent with the suggestion. Language correction was performed.
3. Did the author include positive control (as anti-cancer) in the metabolomics assays? Did they notice the same pathways?
Response: Thank you for your feedback. Originally, the aim of this study was to assess the metabolic alterations induced by the Esperanza extract on K562 cells. Consequently, a metabolomic analysis of cell lysates treated with Doxorubicin was not conducted. We plan to explore this aspect in future studies, possibly in conjunction with other plant extracts and chemotherapeutic agents.
Round 2
Reviewer 1 Report
Comments and Suggestions for Authors
Authors have sufficiently addressed my cooments.
Reviewer 2 Report
Comments and Suggestions for Authors
No further comments
Comments on the Quality of English LanguageModerate langugae editing